

# Comparative analysis of fatty acid metabolism based on transcriptome sequencing of wild and cultivated *Ophiocordyceps sinensis*

Han Zhang, Pan Yue, Xinxin Tong, Tinghui Gao, Ting Peng and Jinlin Guo

Chengdu University of Traditional Chinese Medicine, Chengdu, China

## ABSTRACT

**Background**. *Ophiocordyceps sinensis* is a species endemic to the alpine and high-altitude areas of the Qinghai-Tibet plateau. Although *O. sinensis* has been cultivated since the past few years, whether cultivated *O. sinensis* can completely replace wild *O. sinensis* remains to be determined.

**Methods**. To explore the differences of *O. sinensis* grown in varied environments, we conducted morphological and transcriptomic comparisons between wild and cultivated samples who with the same genetic background.

**Results**. The results of morphological anatomy showed that there were significant differences between wild and cultivated *O. sinensis*, which were caused by different growth environments. Then, a total of 9,360 transcripts were identified using Illumina paired-end sequencing. Differential expression analysis revealed that 73.89% differentially expressed genes (DEGs) were upregulated in *O. sinensis* grown under natural conditions compared with that grown under artificial conditions. Functional enrichment analysis showed that some key DEGs related to fatty acid metabolism, including acyl-CoA dehydrogenase, enoyl-CoA hydratase, 3-ketoacyl-CoA thiolase, and acetyl-CoA acetyltransferase, were upregulated in wild *O. sinensis*. Furthermore, gas chromatography-mass spectrometry results confirmed that the fatty acid content of wild *O. sinensis* was significantly higher than that of cultivated *O. sinensis* and that unsaturated fatty acids accounted for a larger proportion.

**Conclusion**. These results provide a theoretical insight to the molecular regulation mechanism that causes differences between wild and cultivated *O. sinensis* and improving artificial breeding.

Corresponding author
Jinlin Guo, guo596@cdutcm.edu.cn

## INTRODUCTION

*Ophiocordyceps sinensis*, a complex of larval corpses (sclerotia) and fungal progeny formed by infecting Hepialidae insect larvae, has been used as a valuable material in traditional Asian medicine for more than 2000 years (*Sung et al., 2007*; *Lo et al., 2013*). Modern medical research has proven that *O. sinensis* has multiple pharmacological effects such as immune

regulation and antibacterial, anti-tumor, and anti-oxidation effects (*Gao, 2014*; *Liu et al., 2015*; *Dong et al., 2016*).

In recent years, overharvesting owing to global warming and large market demands have led to the rapid reduction of wild *O. sinensis* populations (*Dong et al., 2016*). To alleviate the shortage and high price, the development of the artificial cultivation industry of *O. sinensis* has been greatly promoted. Artificial cultivation ensures the consistency of the genetic background, thereby creating a comfortable growth environment for *O. sinensis*, artificially avoiding abiotic stress, and considerably shortening the growth cycle (*Liu et al., 2016*; *Xiao et al., 2018*). As a result, cultivated *O. sinensis* differs from wild *O. sinensis* in tissue morphology, stress resistance, accumulation of certain biologically active ingredients, etc., which may also cause differences in efficacy. Therefore, whether cultivated *O. sinensis* can completely replace wild populations or have the same medicinal value remains to be determined.

Currently, the comparison between wild and cultivated *O. sinensis* predominantly focuses on the detection of components' contents (*Zheng-Ming et al., 2016*), differences in identification (*Yang et al., 2017*), and research on the pharmacological effects of certain types of biologically active ingredients (*Zhou et al., 2008*); however, such comparisons do not fundamentally explain the formation mechanism of these differences. With the rapid development of high-throughput technologies, applying omics methods to study differences in biological characteristics has become an effective method (*Zhu et al., 2014*; *Xia et al., 2016*; *Xia et al., 2017*). For instance, using mRNA sequencing data of cultivated and wild tomatoes, wild tomatoes were shown to have better salt tolerance and drought resistance based on the molecular levels (*Dai et al., 2017*). Another study compared the transcriptomes of wild and cultivated ginseng and found that HMG-CoA synthase, mevalonate kinase, and squalene epoxidase are associated with ginsenoside biosynthesis. The levels of key enzymes of ginseng are upregulated in wild ginseng, which explains the lower ginsenoside content in cultivated ginseng (*Zhen et al., 2015*).

In this study, we obtained morphological differences related to the developmental phenotype of *O. sinensis* by morphological comparison. On the other hand, focused on the whole genome information of *O. sinensis* by comparing the transcriptome sequencing data of wild and cultivated specimens, mining the key genes that led to differences, and finally verifying the results using quantitative real-time polymerase chain reaction (qRT-PCR) and gas chromatography-mass spectrometry. This research has important theoretical significance and application value for perfecting the molecular regulation mechanism that causes differences between wild and cultivated *O. sinensis* and improving artificial breeding.

## MATERIALS & METHODS

### Specimen collection

Naturally growing *O. sinensis* were harvested at Aba Prefecture, Sichuan Province, China (4,500 m, N31°08′51.9″, E102°21′26.78″) in May. Cultivated samples were collected from the artificial cultivation workshop at Chengdu Eastern Sunshine Co. Ltd. The genetic

background of *Hepialidae* larvae and *H. sinensis* used in the cultivation is consistent with the wild genetic background. Cultured conditions: one month after the artificial infestation has successfully formed sclerotium, keeping light for 10 h a day, air humidity above 80%, soil humidity above 20%, temperature18 °C, and adding nutrients to the soil according to the proportion of the large elements of MS medium. All samples were stored at −80 °C until further processing.

## Morphological comparison

To identify morphological differences between the samples, we first compared the appearance of wild and cultivated *O. sinensis* and recorded photographs. Subsequently, crosscutting was performed at the top and tail of the worm body and the fruiting body at one cm, including the middle part, and photographed using Toup View imaging system. Finally, microscopic analysis was performed to further observe the microstructure characteristics, (*Jie-Pin, 2006*). Paraffin sections were prepared and observed under an electron microscope. Three replicates were conducted for each set of samples.

## Transcriptome data acquisition

Total RNA was extracted using Eastep Super Total RNA Extraction Kit (Promega Shanghai) according to the manufacturer's instructions. After assessing the RNA quality, 1 μg of total RNA from each group was used as input material for RNA sample preparation using the NEBNext$^R$ Ultra$^{TM}$ Directional RNA Library Prep Kit for Illumina$^R$ (NEB, USA) to construct cDNA libraries. Pair-end sequencing was performed using the Illumina Hiseq2000 platform. A total of 100-base paired-end clean reads were generated after removing low-quality reads, adaptor-polluted reads, and reads with high levels of an unknown base. Two biological replicates were performed for RNA-seq.

## Gene expression analysis

Raw data were preprocessed using the fastp tool (http://github.com/openGene/fastp) to obtain clean reads. The Tophat2 aligner tool (http://ccb.jhu.edu/software/tophat/index.shtml) was then used to align sequencing reads to the reference sequences (*Daehwan et al., 2013*). To eliminate the effect of different sequencing discrepancies and gene lengths on the gene expression calculation, fragments per kilobases of exon per million fragments mapped (FPKM) values were used to normalize the expression level (*Robinson & Oshlack, 2010*). To infer the transcriptional changes in both groups, differential expression analysis was performed using the DESeq2 R package according to the express count matrix (https://bioconductor.org/packages/release/bioc/html/DESeq2.html) (*Love, Huber & Anders, 2014*). A false discovery rate (FDR) ≤0.01 and |log2(fold change, FC)|≥2 were set as thresholds for the selection of differentially expressed genes (DEGs).

## Functional annotation and enrichment of DEGs

All DEGs were annotated using five databases, including the NCBI nonredundant protein database, the Kyoto Encyclopedia of Genes and Genomes (KEGG), Universal Protein, and the Cluster of Orthologous Groups of proteins (COG). Gene Ontology (GO) annotations were classified into three ontology categories: molecular function, cellular component, and

biological process (*Ashburner et al., 2000*). KEGG enrichment analysis of the DEGs was performed using the KOBAS software (http://kobas.cbi.pku.edu.cn/kobas3/) (*Minoru & Susumu, 2000*) and showed the top 20 pathways with the smallest significant Q value.

## qRT-PCR

qRT-PCR was performed on the CFX ConnectTM Optics Module using 2X Ultra SYBR Mixture (TransGen, Beijing, China) according to the manufacturer's instructions. The $\beta$-actin gene was used as a reference to normalize the expression data, and the $2^{-\Delta\Delta}$ values were shown as relative expression levels (*Thomas, Kenneth & Livak, 2008*). All primers used for the assessment of transcript levels are listed in Table S1.

## Fatty acid extraction and identification using GC-MS

We collected three batches of wild and cultivated *O. sinensis* specimens for fatty acid testing (Table S2). The samples were prepared dried at 60 °C and grounded to obtain 80~100 mg of powder. two mL 5% hydrochloric acid solution-methanol (1 :1, V/V) and three mL chloroform-methanol were added, then water bathed at 85 °C for 1 h. Next, when cooled to room temperature, one mL of n-hexane was mixed and extracted for 2 min. 100 μL of the supernatant was taken into a one mL volumetric flask and n-hexane was used to make the volume constant. Finally, the fatty acid preparation sample was obtained by filtration through a 0.45 μm microporous membrane (*Zhihui et al., 2017*). All the extracts were analyzed using an Agilent 7890B/5977B gas chromatograph. A TG-5MS capillary column (30 m × 0.25 mm × 0.25 μm) was used with helium (99.999%) as the carrier gas at a constant flow rate of 1.2 mL min$^{-1}$. The sample (0.1 μL) was injected in the split-less mode, and the inlet temperature was 290 °C. The oven temperature program was as follows: the initial temperature was maintained at 80 °C for 1 min, increased to 200 °C at the rate of 10 °C/min, increased from 200 °C to 250 °C at the rate of 10 °C/min, and maintained at 250 °C for 3 min. Finally, the temperature was increased to 270 °C at the rate of 2 °C/min. The qualitative and quantitative analysis of fatty acids was performed according to the GC-MS NIST Mass Spectral Library and the chromatogram of mixed acid fatty standard (NU-CHEK, USA, M20-D, 25mg).

## Data availability

The *O. sinensis* reference genome was derived from the Ensembl Fungi database, ASM44836v1 (http://fungi.ensembl.org/Ophiocordyceps_sinensis_co18_gca_000448365/Info/Index). The raw data of wild *O. sinensis* were deposited in the Genome Sequence Archive under the accession number PRJCA000970, and the data of cultivated *O. sinensis* are available at the NCBI Sequence Read Archive under accession numbers SRR5282577 and SRR5282578.

# RESULTS

## Morphological comparison between wild and cultivated *O. sinensis*

To perform a preliminary comparison between wild and cultivated *O. sinensis*, the physical characteristics, cross-sectional characteristics of each part, and microscopic observations

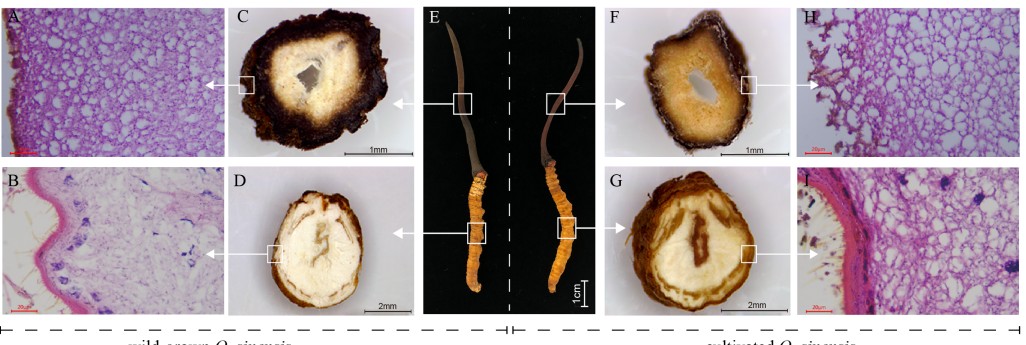

wild-grown *O. sinensis*     cultivated *O. sinensis*

**Figure 1   Overview of the structure and morphology of wild-grown and cultivated *O. sinensis*.** (A–B, H–I), The microscopic characteristics of fruiting body and worm (middle position of structure); (C–D, F–G), The anatomical section characteristics of fruiting body and worm (middle position of structure). (E) The characteristics of appearance of *O. sinensis*.

of paraffin sections were evaluated to determine the morphological differences associated with the developmental phenotype of *O. sinensis*. The structural characteristics of wild and cultivated *O. sinensis* are consistent with the description in the Chinese Pharmacopoeia: "The insect body is connected to the fungus seat, the surface is dark yellow to yellowish brown, and eight pairs of feet" (Fig. S1), and the main difference is reflected in the morphology. The cross-section and microscopic results of each part are shown in Fig. 1, Figs. S2 and S3. Compared with cultivated *O. sinensis*, the outer wall structure is thinner, and the digestive tract and other organs in the body are less complete. In addition, the inner hyphae are arranged in parallel in the fruiting body, the hyphae in the wild specimens appear to be arranged more closely, and the density is significantly higher than that in cultivated *O. sinensis* specimens. Such differences are mainly caused by their physiological functions and growth environment.

## Overview of transcriptome and differential analysis between wild and cultivated *O. sinensis*

After performing quality control of the sequencing data, a total of 28.85 Gb of clean data was obtained, and the percentage of Q30 bases was ≥93.68% (Table S3), indicating that the higher the accuracy of base identification during the sequencing process, the sequence quality can be used for subsequent analysis. The efficiency of comparison with the reference genome was between 81.00% and 84.40% (Table S4), and 9360 transcripts were obtained after mapping, including 997 newly predicted genes.

DEGs were calculated based on FPKM, with an FDR of <0.01 and |log2(fold change, FC)| ≥2. A total of 563 DEGs in the wild and cultivated *O. sinensis* were identified. Compared with the wild group, the expression of 415 genes was downregulated and the expression of only 148 genes was upregulated in the cultivated group. The number of DEGs in the wild group was significantly higher (Fig. 2).

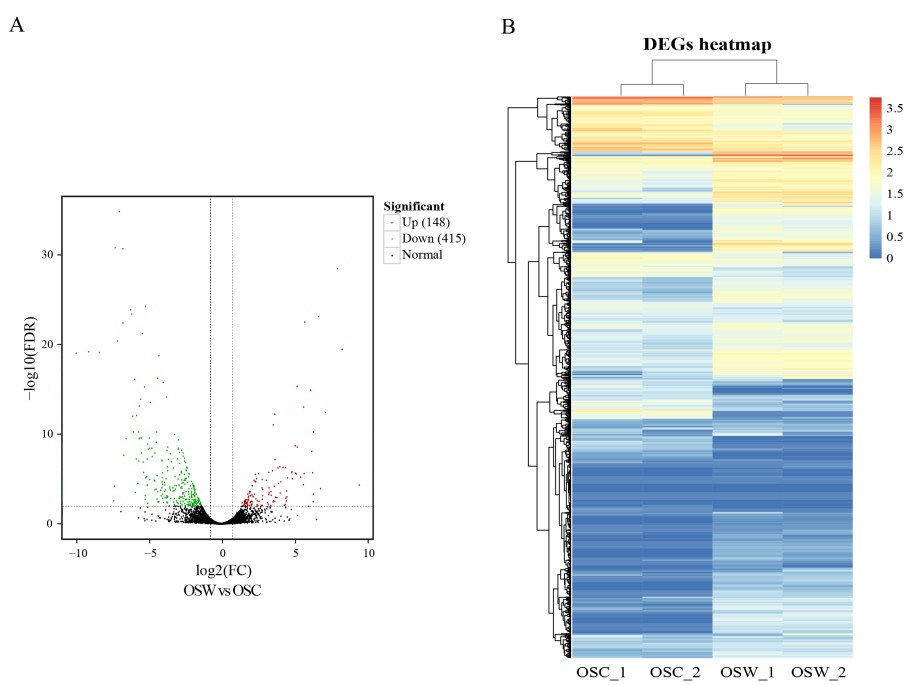

**Figure 2** **Results of differential analysis.** (A) Volcano map of differentially expressed genes (wild vs. cultivated). (B) Heatmap of differentially expressed genes.

## Functional enrichment analysis of DEGs

Overall, 563 DEGs were annotated by Blast2GO to a total of 274 annotation entries. The GO term cellular component was mainly enriched into 14 terms, including membrane, organelle, macromolecular complex, etc. In the GO term molecular function, a total of 225 DEGs were enriched into 8 terms, mainly catalytic activity, binding, transcription regulator activity and transporter activity, etc. In biological process, 193 DEGs were significantly enriched in 14 terms, mainly metabolic process, cellular process, and single-organism process (Fig. 3A). KEGG annotation and enrichment results are shown in Table S5 and Fig. 3B. Fatty acid degradation, fatty acid metabolism, butanoate metabolism, and $\beta$-alanine metabolism showed the most significant differences in the metabolic pathways, of which the most abundant was biosynthesis of antibiotics.

Based on these results, we found that fatty acid metabolism was the most significant difference between wild and cultivated *O. sinensis*. On comparing these data with major functional databases, the DEGs in the fatty acid pathway were identified and were found to code for acyl-CoA dehydrogenase (ACAD) (OCS_02728), enoyl-CoA hydratase (ECH) (OCS_06591), 3-ketoacyl-CoA thiolase (KAT) (OCS_04932), and acetyl-CoA acetyltransferase (ACAT) (OCS_01242/OCS_05099).

## Fatty acid metabolism-related DEGs involved in wild and cultivated *O. sinensis*

Based on the results of functional enrichment analysis of DEGs, we selected fatty acids as the focus of research on the differences between wild and cultivated *O. sinensis*. Acetyl-CoA
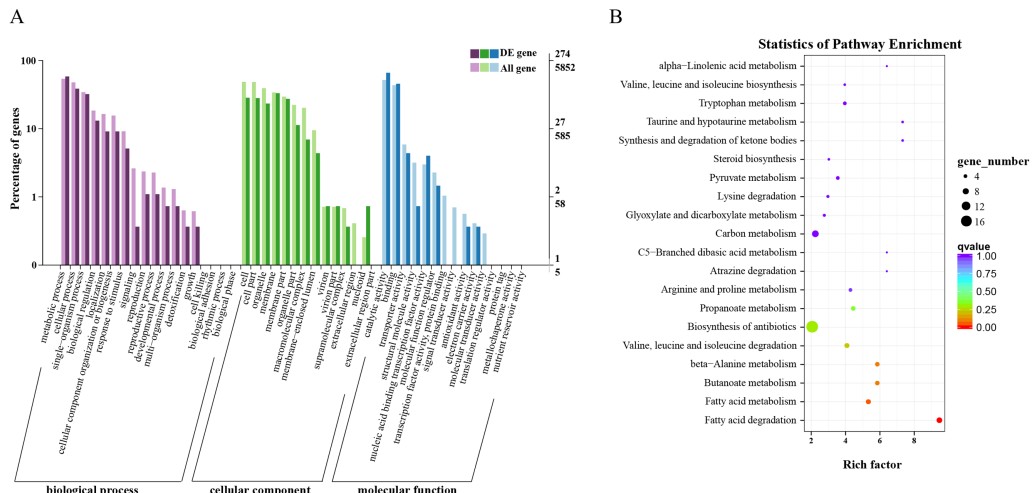

**Figure 3 Functional annotation and enrichment results of differentially expressed genes (DEGs).** (A) Gene ontology functional enrichment map of DEGs. (B) Analysis of Kyoto Encyclopedia of Genes and Genomes significant enrichment of DEGs.

(Ac-CoA) is the precursor of the Malongl-CoA (MalCoA) in fungi and is the final product of fatty acid degradation. Under the catalysis of fatty acid synthase (FANS) and other enzymes, one molecule of MalACP is added in each step, increasing the length of two carbon atoms each time; after seven additions, hexadecanoyl-[acp] is formed and, finally, palmitic acid is formed under the action of FASN (Fig. 4A) (*Nowinski et al., 2018*). In addition, hexadecanoyl-CoA can be used as a substrate for the synthesis of longer-chain fatty acids, generating different types of fatty acids under the catalysis of different enzymes.

In this study, the enzymes encoded by the DEGs of the significantly enriched metabolic pathways participated in the *O. sinensis* fatty acid $\beta$-oxidation pathway, including four steps: dehydrogenation, hydration, re-dehydrogenation, and thiolysis (*Shen & Burger, 2009*). ACAD, ECH, and KAT were significantly downregulated in cultivated *O. sinensis*. ACAD participates in the first reaction of fatty acid and amino acid $\beta$-oxidation under the action of flavin adenine dinucleotide cofactor (*Kim & Miura, 2004*). ECH catalyzes the second step of fatty acid $\beta$-oxidation, promoting the cis-addition of water molecules on the trans-2-enoyl-CoA thioester double bond to form $\beta$-hydroxyacyl-CoA thioester (*Willadsen & Eggerer, 2008*). KAT catalyzes the final step of fatty acid $\beta$-oxidation to produce one molecule of Ac-CoA and one molecule of acyl-CoA (*Mano et al., 1996*). The $\beta$-oxidation that occurs in mitochondria and peroxisomes is the most important form of oxidation and provides a large amount of energy for organisms by coupling with the tricarboxylic acid cycle and electronic respiratory chain (*Neely & Morgan, 1974*). The results showed that the genes encoding various enzymes involved in the degradation and synthesis of fatty acids were downregulated in cultivated *O. sinensis*, which indicates that fatty acid metabolism is more active in wild *O. sinensis* (Fig. 4B). Moreover, the product of palmitic acid decomposition is Ac-CoA. When fatty acid metabolism is weakened, the downstream citric acid cycle/tricarboxylic acid cycle, alanine aspartate metabolism, and
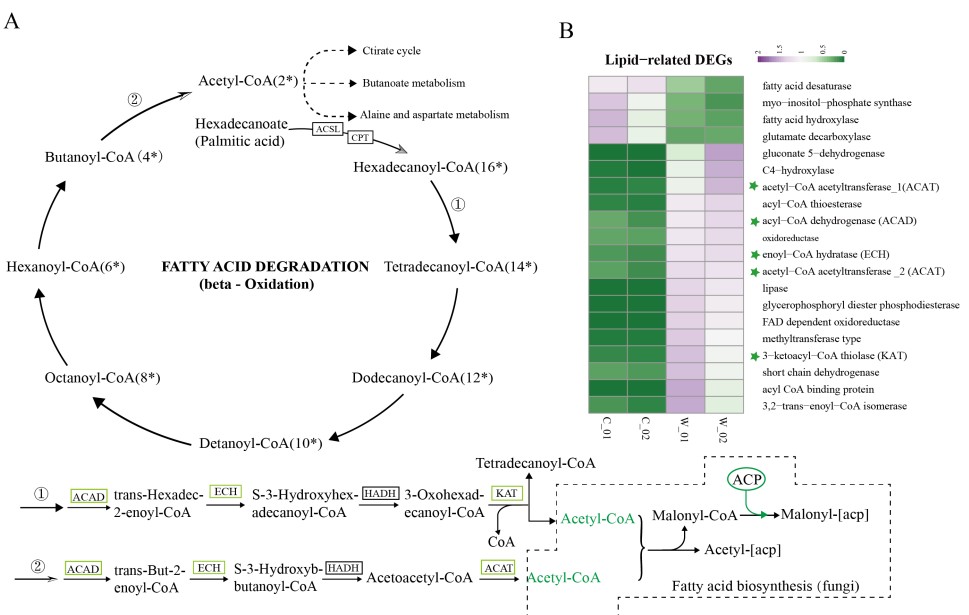

**Figure 4** **Metabolism and degradation of palmitic acid in *O.sinesis*.** (A) Degradation of palmitic acid in *O.sinesis*. (1) Palmitoyl coenzyme A is oxidized and decomposed to generate myristoyl-CoA. (2) Butyryl-CoA is oxidized and decomposed to produce Acetyl-CoA. (B) Heatmap of lipid-related DEGs, the green marked as a key gene in palmitic acid degradation pathway. *ACSL, long-chain acyl-CoA synthetase. CPT, carnitine O-palmitoyltransferase. ACP, acyl carrier protein.

butyrate metabolism also weakens (Fig. 4A). Because of the long growth cycle of wild *O. sinensis* and the lack of nutrients in the living environment, we speculate that *O. sinensis* relies mainly on its own fatty acid oxidative decomposition to provide energy for itself, and on $\beta$-oxidation, especially via palmitic acid decomposition.

## Validation of transcriptome data by qRT-RCR
To confirm the reliability of the sequencing data, 12 DEGs were randomly selected to validate the RNA-Seq expression profiles. As expected, qRT-PCR results showed that most of the mRNAs shared similar expression with those from the sequencing data. The expression levels of ten genes were downregulated in the cultivated *O. sinensis* (Fig. 5) and those of two genes were upregulated, indicating that our data were accurate and reliable in the subsequent analyses.

## Quantitative analysis of fatty acid content
GC-MS was performed to analyze the fatty acids of *O. sinensis*. The results showed that the types of fatty acids in the wild and cultivated specimens were basically the same (Fig. 6) but that the content was quite different. The content of fatty acids in the wild group was much higher than that in the cultivated group, which is consistent with the previously reported (*Shen & Burger, 2009*). Furthermore, among all types of fatty acids, oleic acid (C18:1n9c), linoleic acid (C18:2n6c), and palmitic acid (C16:0) are the most important components and account for up to 89.38%–94.88%. From the perspective of fatty acid

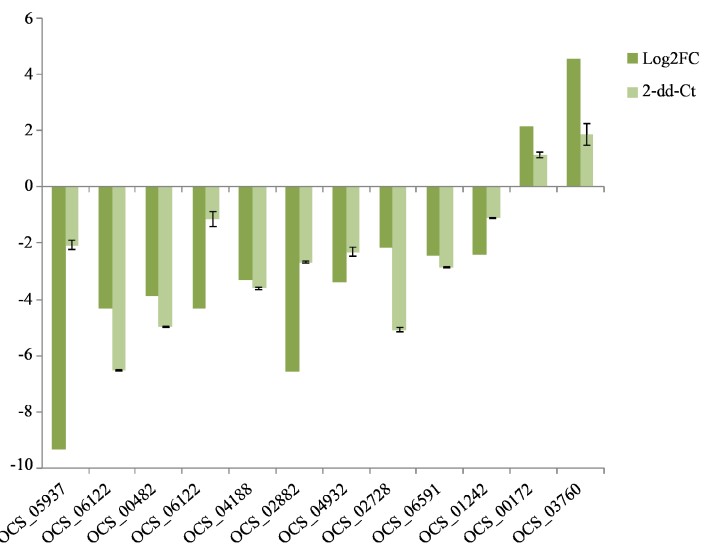

**Figure 5** **QRT-PCR verification of the expressed genes in Illumina sequencing.** The expression value of $\beta$-actin was used as an inner control, the wild as a control compared to cultivated. All data are means of three replicates with error bars indicating SD.

types, the content of unsaturated fatty acids in *O. sinensis* was higher than the content of saturated fatty acids. In addition, the content of fatty acids in *O. sinensis* has a certain aging effect. With the extension of storage life, the content of fatty acids in medicinal materials is continuously reduced (Table S6).

## DISCUSSION

It is well known that *O. sinensis* is a typical entomogenous fungus. When *Hirsutella sinensis* successfully infects the host Hepialus larvae, its bud tube breaks to form a long spindle-shaped fungus, which proliferates and differentiates in an apical budding manner and matches with each other. The phytoplasma grows hyphae and gradually fills the entire body to form sclerotia (*Zhang et al., 2011*; *Zhang et al., 2012*; *Guo et al., 2015*). Compared with *O. sinensis* growing in a comfortable artificial environment, wild *O. sinensis* cannot absorb more nutrients from the environment and can only rely on the insect body itself, which causes a greater degree of decomposition in an insect body part and a looser texture.

In the Qinghai-Tibet Plateau, which is subject to low temperatures and strong ultraviolet rays, the main vegetation type is the alpine meadow area, and the texture is sandy loam and harder soils (*Zhou et al., 2018*). These environmental factors directly affect the formation of sclerotia and fruiting bodies of *O. sinensis*. For instance, under natural conditions, the sclerotium needs to overcome the strong resistance of the soil when it grows out of the pedestal; thus, the sclerotium is stronger and tougher than that growing in good conditions and has a tighter organization. In addition, the length and intensity of light received by wild *O. sinensis* is greater, resulting in slower growth but stronger fruiting bodies (larger diameter stroma). In contrast, the fruiting bodies grow rapidly and slender without this condition, similar to cultivated *O. sinensis* (*Tu et al., 2010*; *Tong et al., 2020*).

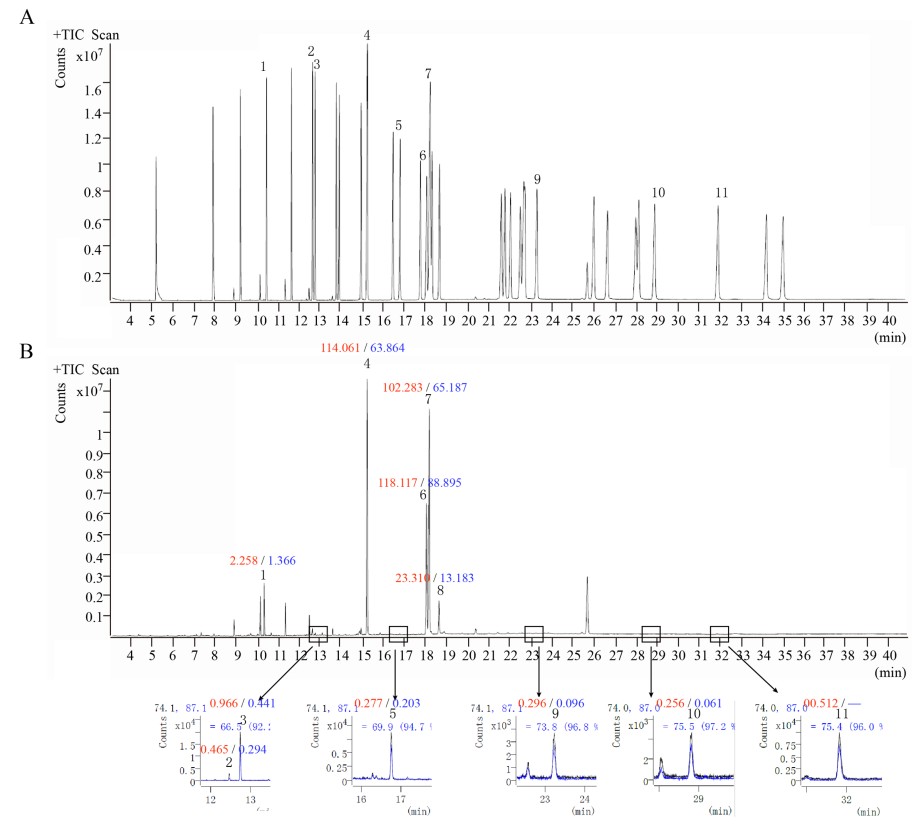

**Figure 6  The ion chromatograms of fatty acid.** (A) Total ion chromatograms of mixed fatty acid methyl ester standard. (B) The ion chromatograms of *O. sinensis* (OSW_S1). 1-Methyl dodecanoate (C12:0); 2-Methyl myristoleate (C14:1n5); 3-Methyl myristate (C14:0); 4-Methyl palmitate (C16:0); 5-Methyl heptadecanoate (C17:0); 6-Methyl linoleate (C18:2n6c); 7-Methyl oleate (C18:1n9c); 8-Methyl stearate (C18:0); 9-Methyl arachidate (C20:0); 10-Methyl dodecanoate (C22:0); and 11-Methyl tricosanoate (C23:0). Red numbers represent the content (mg/g) in the wild group, blue numbers represent the content in the cultivated group, "-" indicates content below the detection limit.

In this study, the genetic background of wild and cultivated *O. sinensis* was identical, but there was a large difference in the growth environment. Because the wild *O. sinensis* need to resist the harsh environment of the plateau, the DEGs identified in this study may play an important role in the process of growth and metabolism, differences in substance accumulation, and response to abiotic stress. Studies have reported that the types and contents of chemical components are not much different between wild and cultivated *O. sinensis* (*Hongxia, Xinhua & Wei, 2012*; *Liu et al., 2016*; *Zheng-Ming et al., 2016*). However, the palmitic acid and oleic acid content of the fatty acids, which are the main components, is much higher in wild than in cultivated *O. sinensis* (*Shen & Burger, 2009*). This conclusion is consistent with our transcriptome data analysis results, which can be an indicator to distinguish between wild and cultivated *O. sinensis*.

At present, research on fatty acids in *O. sinensis* mainly focuses on detection methods and composition comparisons (*Yuan-Can et al., 2015*), and only few studies have assessed the physiological functions. It was proved that high cold hardiness plants have a

higher proportion of unsaturated fatty acids, and a lower phase transition temperature can maintain the fluidity of the membrane at low temperatures to maintain normal physiological functions (*Šajbidor, 1997*). Research on the cold resistance of winter wheat found that when the temperature decreases, the content of unsaturated linolenic acid increases and that of palmitic acid decreases (*Dong-Wei et al., 2013*), indicating that these two fatty acids play the greatest role in the cold resistance of winter wheat. Therefore, it is speculated that linolenic acid and palmitic acid can be used as indicator fatty acids for cold resistance in winter wheat. In fact, wild *O. sinensis* is subject to long-term low-temperature stress, which may be attributed to a significantly higher content of unsaturated fatty acids than cultivated *O. sinensis*.

## CONCLUSION

In summary, a comparative morphological and transcriptomic analysis of *O. sinensis* grown in varied environment was performed. The present results show that natural and cultivated environments will cause significant differences in the morphology. Reflected in the high density of wild *O. sinensis* fruiting body hyphae and higher digestibility of the worm body. According to the transcriptome, genes involved in fatty acid metabolism were upregulated including ACAD, ECH, KAT and ACAT, which conformed that fatty acids are higher in wild *O. sinensis* at the molecular level. Furthermore, this research has important theoretical significance and application value for perfecting the molecular regulation mechanism that causes differences between wild and cultivated *O. sinensis* and improving artificial breeding.

### Funding
This work was supported by the National Natural Science Foundation of China (No. 81872959, 81373920, 30801522) and the Sichuan Province Youth Innovation Team Fund (No. 19CXTD0055). The funders had no role in study design, data collection and analysis, decision to publish, or preparation of the manuscript.

### Grant Disclosures
The following grant information was disclosed by the authors:
National Natural Science Foundation of China: 81872959, 81373920, 30801522.
Sichuan Province Youth Innovation Team Fund: 19CXTD0055.

### Competing Interests
The authors declare there are no competing interests.

### Author Contributions
- Han Zhang conceived and designed the experiments, performed the experiments, analyzed the data, prepared figures and/or tables, authored or reviewed drafts of the paper, and approved the final draft.

- Pan Yue performed the experiments, analyzed the data, prepared figures and/or tables, and approved the final draft.
- Xinxin Tong and Jinlin Guo conceived and designed the experiments, authored or reviewed drafts of the paper, and approved the final draft.
- Tinghui Gao and Ting Peng performed the experiments, prepared figures and/or tables, and approved the final draft.

## DNA Deposition

The following information was supplied regarding the deposition of DNA sequences:

The data is available at Genome Sequence Archive: PRJCA000970 / CRP000595; NCBI Sequence Read Archive: SRR5282577, SRR5282578.

https://ngdc.cncb.ac.cn/gsa/browse/CRA001047.

## Data Availability

The *O. sinensis* reference genome was derived from the Ensembl Fungi database: ASM44836v1.

The PCR data are available as a Supplemental File.

http://fungi.ensembl.org/Ophiocordyceps_sinensis_co18_gca_000448365/Info/Index.

## Supplemental Information

Supplemental information for this article can be found online at http://dx.doi.org/10.7717/peerj.11681#supplemental-information.

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
