# Peer review of "Comparative analysis of fatty acid metabolism based on transcriptome sequencing of wild and cultivated Ophiocordyceps sinensis"

_PeerJ, doi:10.7717/peerj.11681_

## Round 0.1 · original submission · Major Revisions

I have received two reports from reviewers with complementing expertise. Both reviewers find your study of some interest but think that major revisions are required. They make many suggestions on how to improve the manuscript and enhance clarity. Reviewer 1 points out that the discussion and background information on lipid metabolism contains errors and needs in-depth revision together with correcting figure 4. It is necessary that this point is appropriately addressed in a revised manuscript.

The methods lack detail as pointed out by both reviewers. Both reviewers request a detailed summary of the method described in Zhihui et al., 2017 to make it accessible to a non-Chinese speaking audience. It will be absolutely necessary to improve and extend the methods section in a revised version as suggested by the reviewers.

The statistical analysis is poorly developed and needs to be improved as pointed out by reviewer 2.

Both reviewers comment on many typos, spelling errors and odd sentences. Please revise carefully. Make sure that all figures are referenced in the main text.

Although large parts of the manuscript require in-depth revision, no additional experiments are necessary to make the manuscript acceptable.

Reviewer 1 ·

Basic reporting

1. weak. the manuscript contains several spelling mistakes as well as odd sentences which obfuscate than help to explain findings and discussion.
2. I recommend deeper and more focused on fungi metabolism selection of references. All cited references should be checked again for proper and uniform formating. Correct in line 202, 327 reference “GERHARDT”; it should not be capitalicized . I don’t think the publication on fatty acid metabolism in higher plants is a good reference here. One should refer to either fungi or related eukaryota metabolic pathways.
3. structure ok
4. Line 124 - the publications by Zhihui et al., 2017 is not indexed in NCBI, nor in other non-chinese speaking platforms. I would like to see much detailed description of the method as I could not verify approach applied in this study.

Experimental design

proper

Validity of the findings

Line 191-193 – this is not a cycle. Both proceses are spatially and temporarly separated and regulated not to form a futile cycle. Remove!
Line 210-211. Fatty acids are not sources of PL and GL. FA are used to synthesize these lipids, and in turn they can be degraded and release FA. The sentence is false.
Figure 4. Fatty acid oxidation (FAO) takes place in mitochondria or/and in peroxisomes in Ascomycota. There is no consensus about the localization of the pathway in various fungi groups. According to Shen&Burger 2008 Sordariomycota, to which O. Sinensis belongs, utilise mitochondrial and hybrid peroxisomal FAO pathway. As stated before, fatty acids (palmitic acid) synthesis is located in cytosol and it is regulated differentially than FAO. In the bottom of the figure authors suggest that elongation of an acyl chain by two carbons consumes one acetyl-CoA. Wrong. In yeast (and other eukaryotes, thus probably in O. sinensis too), a malonyl-CoA unit is consumed to add 2C. Mitochondria of O. sinensis might possess a conserved pathway of FA synthesis – used to form eight-carbon-long lipoate, a cofactor of KGDH (e.g. Nowinski et al Curr Biol. 2018). The figure 4 suggest that de novo synthesis and FAO coexist and function as opposite processes. This is wrong and needs to corrected.
There are spelling mistakes in this wrong figure.
Line 206 the subclause: ...” and can also be used as a branched-chain amino acid (Ile, Leu, and Val) metabolic enzymes” has little meaning here – remove it. There are dehydrogenases specific for branched amino acids and they are not part of the FAO, so as ACAD does not necessarily takes part in BCAA catabolism.
Line 216-219 These two sentences little sense. Butyrate metabolism wasn’t mentioned before. What is the relevance of butyrate metabolism in O. sinensis? Is this one of key synthesized metabolites of it? How “alanine aspartate metabolism” is related with acetyl-CoA. Is FAO the main mitochondrial acetyl-CoA producing pathway in the fungus? What is the proportion of glycolytic pyruvate or degradation of host proteins (supply of TCA intermediates from amino acids degradation) ? How elevated expression of biosynthetic and catabolic genes would supply more energy to the wild fungus?

Additional comments

Line 99 – invalid URL, is “cdu" should be “edu”
Line 136 – invalid URL
Line 193 – Acetyl-CoA should be abbreviated Ac-CoA. Correct throughout the manuscript.
Line 207 “homeopathic”
Line 213 “electronic”, spelling mistake
Line 233 what are these “previously reported conclusions”?
Line 258 what is meant by the term “strong fruiting bodies”? Clarify.
Line 283-286 rephrase the sentence as it makes little sense at the moment

Reviewer 2 ·

Basic reporting

In the manuscript entitled “Comparative analysis of fatty acid metabolism based on transcriptome sequencing of wild and cultivated Ophiocordyceps sinensis”, the authors aimed to investigate the differences of O. sinenesis by performed morphological and transcriptomic comparisons between wild and cultivated O. sinensis. The manuscript generally includes the appropriate references to describe the background of this study. However, it is needed to inform and describe the specimens collection for further DEGs analysis and statistical analysis. Besides, some figures poorly represented. They are needed to have large improvements.

Experimental design

The authors should describe detailed methods to provide reliability and reproducibility of the analysis. Some points did not relate to each other between methods and results, the suggestions are in the comments. In particular, when you collected specimens from wild samples and cultivate samples, the authors should inform the substrates for cultivation as well e.g., cultivated O. sinesis. Since DEGs will change upon nutrients. Moreover, for natural growing O. sinesis, the authors also need to describe more details.

Validity of the findings

The authors found that key DEGs related to fatty acid metabolism and were upregulated in wild O. sinensis. The expression level of these genes was validated with the qRT-PCR result. Also, the fatty acid content was determined using GC-MS. These results well support each other. The conclusion is stated and linked to the research question.

Additional comments

Major comments
1. Due to environmental factors affect to O. sinensis, the authors should add more details in the specimen collection part, which season that wild O. sinensis were harvested. Also, the condition which was used for cultured samples should be defined, such as medium or host, temperature, light intensity, and humidity.
2. Which part of O. sinesis (worm body or fruiting body) was chosen for total RNA extraction to perform RNA sequencing and qRT-PCR, and for fatty acids extraction to determine the content of fatty acids?
3. To ensure that an international audience can clearly follow experiment. The method of sample preparation and fatty acid extraction should be explained in your manuscript because, in the previous study (Zhihui et al., 2017), it was reported in the Chinese language.
4. Which criteria were used to select the DEGs? In this manuscript, there are both “FDR ≤ 0.05 and log2(FC) ≥ 1”, and “FDR ≤ 0.01 and fold change > 2”. The statistic method are needed to add in manuscript in details.
5. How many biological replicates were performed for RNA sequencing in each group? The authors stated that “three biological replicates were performed for each sample” in the materials and methods section, but the heatmap of DEGs in Figure 2D shows only two data in each group. Also, the raw data of cultivated O. sinensis which were deposited in SRA has only two accession numbers.
6. The authors stated that “DEGs were calculated based on FPKM” (line 164). However, the differential expression analysis using the DESeq2 R package use median of ratios as a count normalization method, not FPKM. These are needed to address in manuscript.
7. The authors didn’t describe about the results related to Figure 3B., while Figure 3C was mentioned but it’s not presented in your manuscript.
8. In Figure 4., the authors presented the metabolism and degradation of palmitic acid in O. sinensis. I also suggest that the expression value of DEGs involved in the palmitic acid biosynthesis pathway should be included in this Figure.
9. In Figure 6B., The total ion chromatograms show only the peaks of the OSW_S1 sample and the content of fatty acids between the wild and cultivated O. sinesis were presented without statistical value. So, I suggest that the data of the content of all detected fatty acids in wild and cultivated O. sinesis should be presented as mean ± S.D. (n = 3 biological replicates in each group). The authors need to add statistics in the results.

---

## Round 0.2 · accepted · Accept

After reading your manuscript, I came to the conclusion that you have addressed the concerns of the reviewers.